

# A quantitative test of the "Ecomorphotype Hypothesis" for fossil true seals (Family Phocidae)

James Patrick Rule[1,2], Gustavo Burin[1] and Travis Park[1,2,3]

[1] Sciences Group, Natural History Museum, London, United Kingdom
[2] School of Biological Sciences, Monash University, Clayton, Victoria, Australia
[3] Sciences, Museums Victoria Research Institute, Museums Victoria, Melbourne, Victoria, Australia

Corresponding author
James Patrick Rule,
jrule.palaeo@gmail.com

## ABSTRACT

The fossil record of true seals (Family Phocidae) is mostly made up of isolated bones, some of which are type specimens. Previous studies have sought to increase referral of non-overlapping and unrelated fossils to these taxa using the 'Ecomorphotype Hypothesis', which stipulates that certain differences in morphology between taxa represent adaptations to differing ecology. On this basis, bulk fossil material could be lumped to a specific ecomorphotype, and then referred to species in that ecomorphotype, even if they are different bones. This qualitative and subjective method has been used often to expand the taxonomy of fossil phocids, but has never been quantitatively tested. We test the proposed ecomorphotypes using morphometric analysis of fossil and extant northern true seal limb bones, specifically principal components analysis and discriminant function analysis. A large amount of morphological overlap between ecomorphotypes, and poor discrimination between them, suggests that the 'Ecomorphotype Hypothesis' is not a valid approach. Further, the analysis failed to assign fossils to ecomorphotypes designated in previous studies, with some fossils from the same taxa being designated as different ecomorphotypes. The failure of this approach suggests that all fossils referred using this method should be considered to have unknown taxonomic status. In light of this, and previous findings that phocid limb bones have limited utility as type specimens, we revise the status of named fossil phocid species. We conclude that the majority of named fossil phocid taxa should be considered *nomina dubia*.

## INTRODUCTION

The taxonomy and systematics of extinct true seals (Family Phocidae) is heavily influenced by their fossil record, which is notoriously incomplete (*Berta, Churchill & Boessenecker, 2018*; *Valenzuela-Toro & Pyenson, 2019*). The majority of extinct species (53%) have been described from isolated postcranial bones (*Valenzuela-Toro & Pyenson, 2019*; *Berta, Churchill & Boessenecker, 2022*), specifically humeri and femora. These bones have limited

taxonomic utility and therefore make comparisons to other fossils and taxa difficult (*Churchill & Uhen, 2019*).

To assist in the referral of isolated phocid fossils to known or new taxa, an 'Ecomorphotype Hypothesis' was proposed by *Koretsky (2001)*. This hypothesis proposed that phocines (northern true seals) occupied specific ecological niches which were related to the morphology of the mandible, humerus, and femur. These morphologies supposedly enable different seal species (with assumed different ecological niches) to occupy the same region. This technique has subsequently been used to lump isolated fossils into five distinct ecomorphs. With this method, fossils from the same (or sometimes different) formation which are completely isolated and unassociated with one another are purportedly referable to the same species and can expand its hypodigm, even if the elements those fossils represent (*e.g.*, a femur) do not overlap at all with the holotype (*e.g.*, a humerus). Since its original description, this 'ecomorphotype hypothesis' (*Koretsky, Alexander & Rahmat, 2020*) has been expanded to devinophocines (extinct stem-phocids), monachines (southern true seals), and cystophorines (an unsupported true seal subfamily, see below).

This technique, which relies on subjective qualitative morphological coding, has been used to refer a multitude of fossils to species (*e.g., Praepusa vindobonensis, Phocanella pumila, Leptophoca proxima*) with otherwise isolated postcranial type specimens (*Koretsky & Grigorescu, 2002*; *Koretsky, 2003*; *Koretsky & Peters, 2008*; *Koretsky & Ray, 2008*; *Koretsky, Ray & Peters, 2012*; *Koretsky & Rahmat, 2013*; *Koretsky, Rahmat & Peters, 2014*; *Koretsky, Peters & Rahmat, 2015*; *Rahmat & Koretsky, 2016*; *Rahmat et al., 2017*; *Rahmat & Koretsky, 2018*; *Hafed et al., 2023*). This occasionally includes fossils other than the mandible, humerus, and femur, or fossil elements that are fragmentary and incomplete. While the referral of fossil Phocidae with the ecomorphotype hypothesis represents a substantial body of work, the validity of this hypothesis has recently fallen into question (*Dewaele, Lambert & Louwye, 2017a*; *Dewaele et al., 2018*; *Churchill & Uhen, 2019*; *Valenzuela-Toro & Pyenson, 2019*; *Rule et al., 2020a*). Specifically, there has been no quantitative test of this method, which is concerning considering the taxonomic identity of a large portion of the referred phocid fossil record hinges on the validity of this 'hypothesis'.

We aim to quantitatively test the validity of the "Ecomorphotype Hypothesis" for the first time, using morphometric analysis of northern true seal (subfamily Phocinae) limb bones. In addition, we provide a review of the taxonomic status of genera and species of fossil true seals.

## MATERIALS AND METHODS

We used a subset of the dataset published in *Churchill & Uhen (2019)*, which was a measurement protocol of the humeri and femora of phocid seals. The definition for measurements used can be found in *Churchill & Uhen (2019)* and in Fig. S1. As the original Ecomorphotype Hypothesis was defined for Phocinae (*Koretsky, 2001*; *Koretsky, Alexander & Rahmat, 2020*), we only used measurements for fossil and extant Phocinae. We excluded *Cystophora cristata* (hooded seal) as the authors of the Ecomorphotype Hypothesis

consider it to be in a separate subfamily (Cystophorinae; *Koretsky, Alexander & Rahmat, 2020*); however, we note that this assignment contradicts the vast majority of morphological and molecular evidence which supports this taxon as a member of Phocinae (*Fulton & Strobeck, 2010*; *Dewaele et al., 2017b*; *Berta, Churchill & Boessenecker, 2018*; *Paterson et al., 2020*; *Rule et al., 2020b*; *Park et al., 2024*). We then expanded on this dataset using 36 specimens from the Natural History Museum in London (NHMUK) and a specimen of *Phoca largha* (spotted seal) in the National Science Museum in Tokyo (NMNS). Measurements were taken directly from the specimens with digital callipers by a single researcher and were recorded in millimetres, and right and left sides (when both existed) were averaged. The raw data was then transformed so that all other measurements for the humerus were scaled against total humerus length (TLH), and all other measurements for the femur were scaled against maximum length of the femur (MLF). Data was analysed at the specimen level, rather than averaging by species (which would result in few data points). We analysed the data for the humeri and femora both separately and together. Fossils were excluded when humeri and femora were analysed together. For all analyses we used the first four phocine ecomorphotypes defined in *Koretsky, Alexander & Rahmat (2020)*, but restricted the codings to extant taxa and treated fossil taxa as unknowns; because of this, ecomorphotype group 5 was excluded as it is solely consisted of the fossil taxon *Cryptophoca maeotica*.

For the four ecomorphotypes analysed, we follow the definitions as defined by *Koretsky (2001)* and *Koretsky, Alexander & Rahmat (2020)*, and the full definitions of these ecomorphotypes can be found in these publications. These ecomorphotypes are defined by both extant and extinct species that are grouped together based on morphological similarities of the mandible, humerus, and femur, as well as shared dietary characteristics, foraging behaviour, and foraging depths. However, these groupings are problematic due to vague and subjective morphological discriminators (*e.g.*, large *vs.* small) that overlap between groups, inaccurate foraging behaviours (*e.g., Erignathus barbatus*, the bearded seal, does not wear its teeth from crushing shells, but from suction-related wear; *Marx et al., 2023*), and outdated diving depth data (*e.g., Erignathus barbatus* has been recorded deeper than the 100 m limit for Ecomorphotype 1, *Schreer & Kovacs, 1997*). Here we list the extant species included in these groupings and the humeri and femora characters used to validate these groupings.

Ecomorphotype 1 includes the species *Erignathus barbatus*. It is defined by humeri with large lesser tubercles level with the humeral head, shallow intertubercle groove, and a large deltoid crest; and femora with an enlarged greater trochanter higher than the femoral head, and a developed intertrochanter crest that is lower than the femoral head. *Koretsky, Alexander & Rahmat (2020)* states this group typically has worn postcanines from benthic feeding on hard-shelled invertebrate prey, and forage at a depth of 60–100 m (but see commentary above).

Ecomorphotype 2 includes *Phoca vitulina* (harbour seal), *Pagophilius groenlandicus* (harp seal), *Pusa hispida* (ringed seal), *Pusa sibirica* (lake Baikal seal), and *Pusa capsica* (Caspian seal). It is defined by humeri with lesser tubercles higher than the head, a broad
(but shallow) intertubercular groove, and an enlarged proximal width of the deltoid crest; and femora with a greater trochanter higher than the head, a proximal end wider than the distal end, and a poorly developed trochanteric crest. *Koretsky, Alexander & Rahmat (2020)* states that the group is composed of piscovorous/crustacean-feeding predators, which foraging on schools of fish at shallow depths (up to 90 m), with less wear on the postcanines than group 1.

Ecomorphotype 3 includes *Histriophoca fasciata* (ribbon seal). It is defined by lesser tubercles that are slightly higher than the humeral head, a narrowed intertubercular groove, and the deltoid crest being widest either in the middle or at the proximal end; for the femora, the greater trochanter is slightly higher than the femoral head, with the proximal end bevel-shaped, and a short intertrochanteric crest below the trochanteric fossa. *Koretsky, Alexander & Rahmat (2020)* states that the group feeds on various invertebrate prey including amphipods, occasionally fish, and forages near the ocean bottom at 50–100 m deep, and notes that the postcanines are "better adapted" to eating hard shelled prey than group 1 (but these adaptations are not elaborated on). Of note, the amphipod specialist *Pusa sibirica* (*Watanabe, Baranov & Miyazaki, 2020*) is not assigned to group 3 (it is assigned to group 2).

Ecomorphotype 4 includes *Phoca largha* (spotted seal) and *Halichoerus grypus* (grey seal). It is defined by humeri with a rounded lesser tubercle that is higher than the humeral head, small and shallow intertubercular groove, and a deltoid crest that is of equal width along its proximo-distal length; femoral characters defining this group include a greater trochanter higher than the femoral head, a proximal end wider than the distal end, and an intertrochanteric crest that extends to the femoral head which is transverse to the proximo-distal axis. *Koretsky, Alexander & Rahmat (2020)* states that this group consumes mainly pelagic fishes, foraging between 150–300 m depth.

We assessed the utility of the ecomorphotypes of *Koretsky, Alexander & Rahmat (2020)* for the referral of dissociated fossils to specific taxa using several methods. Analyses were performed in R version 4.2.1 (*R Core Team, 2023*), and mostly follow those in *Churchill & Uhen (2019)*. To visualise the morphological variation between the ecomorphotypes, we performed a principal components analysis (PCA) on the datasets using the R function 'prcomp' from the R Stats package (*R Core Team, 2023*). We evaluated the first two principal components only. To determine if ecomorphotypes were a good system to discriminate fossils into separate taxonomic bins, we performed a discriminant function analysis (DFA) on the datasets using the 'lda' function in the R package MASS (*Venables & Ripley, 2002*). We assessed the DFA for the three variations of the dataset using jackknife resampling. We plotted Linear Discriminant 1 (LD1), which explained the greatest variation. The accuracy of the assignment of fossils by the DFA to ecomorphotypes was assessed using the posterior probability.

If the Ecomorphotype Hypothesis is valid, then it is expected that the PCA would demonstrate clear separation between the ecomorphotypes along multiple PC axes, especially the first few as they would explain most morphological variation between groups. In addition, the DFA of the datasets would result in accurate posterior

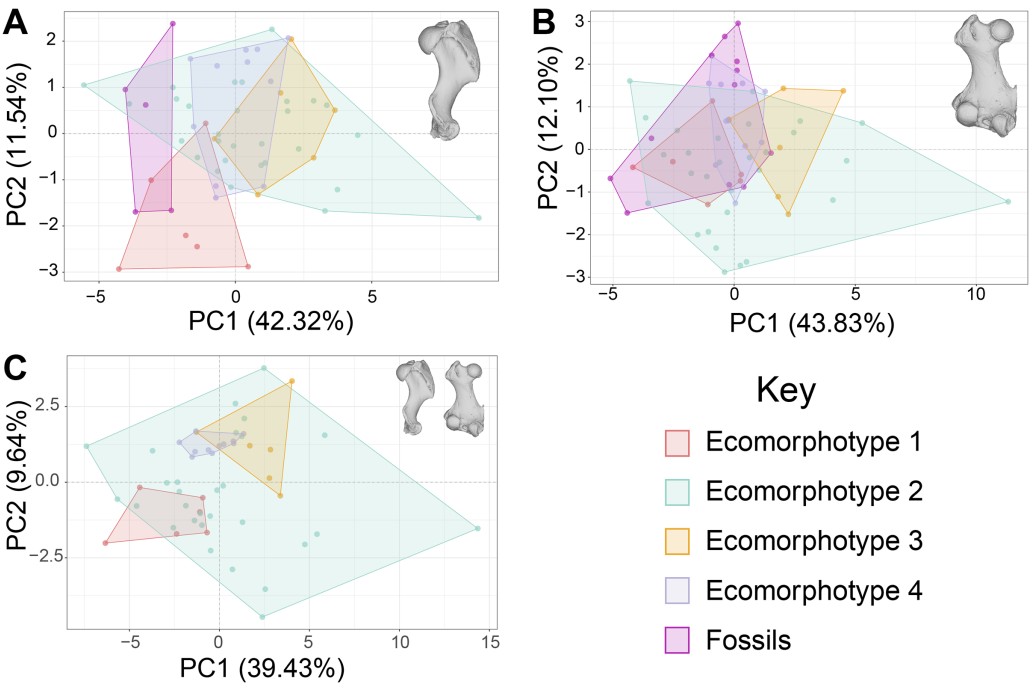

**Figure 1 Principal components analysis (PCA) of phocine postcrania.** Humeri (A), femora (B), and both humeri and femora together (C). The first two PC axes are displayed, and convex hulls are grouped by ecomorphotypes and fossil specimens. Ecomorphotype 1 $n = 6$, Ecomorphotype 2 $n = 29$, Ecomorphotype 3 $n = 6$, Ecomorphotype 4 $n = 11$, Humeri fossils $n = 5$, Femora fossils $n = 12$. Percentages represent amount of variation explained by each PC axis.

probabilities, and the DFA would be able to confidently assign isolated fossils to the ecomorphotypes previously hypothesized using the Ecomorphotype Hypothesis.

## RESULTS

For the principal components analysis: PC1 for the humeri-only dataset represented mostly variation in measurements from the distal articular surface, while PC2 represented variation in the length from the head to the deltopectoral crest, and the depth of the coronoid fossa (Fig. 1A); PC1 for the femur-only dataset represented variation in the vast majority of measurements of the distal articular surface, while PC2 represents variation in the length of the greater trochanter, diameter of the neck, and length of the lateral side of the femur (Fig. 1B); PC1 of the combined dataset represents variation in the distance across the distal condyles of the femur, and the width of the medial condyle of the femur, while PC2 represents variation in the maximum width of the distal epiphysis of the humerus, the distance between the distal condyles of the femur, and the width of the diaphysis of the femur (Fig. 1C).

The PCA of phocine limb bones (Fig. 1) demonstrated substantial overlap between the four ecomorphotypes along both principal components 1 and 2. The exception is a small amount of separation between Ecomorphotypes 1 and 4 on PC2 of the combined dataset (Fig. 1C). In all PCA's, Ecomorphotype 2 occupied the morphospace the most, likely due to the large amount of taxa assigned to this group by *Koretsky, Alexander & Rahmat (2020)*.

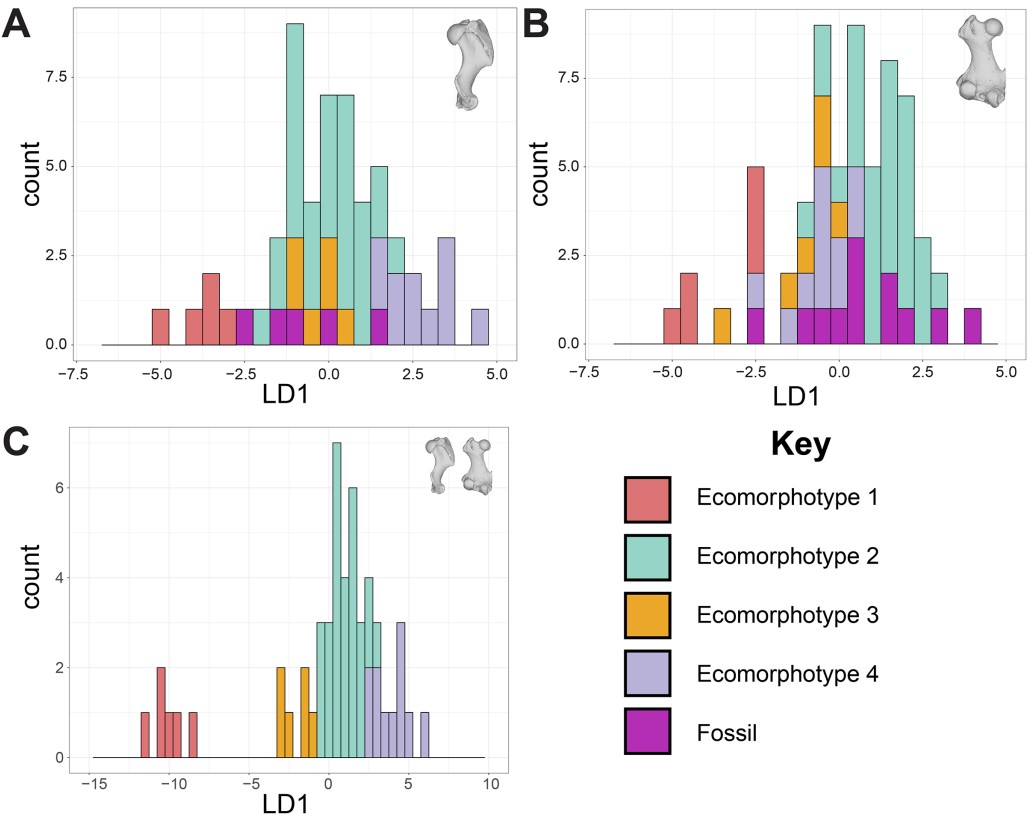

**Figure 2 Discriminant function analysis (DFA) of phocine postcrania.** Humeri (A), femora (B), both humeri and femora (C). Linear Discriminant 1 (LD1) is shown. Proportion of trace captured by LD1: 67.97% (A), 54.1% (B), 71.08% (C). Jackknife validation of Discriminate Function Analyses: 0.69 (A), 0.77 (B), 0.67 (C). The only distinct separation from the Discriminate Function is between Ecomorphotype 1 and all other Ecomorphotypes in the combined dataset.

The principal components (PCs) making up 95% of variation were as follows: the first 11 PCs for the humeri dataset, the first 11 PCs for the femora dataset, and the first 17 PCs for the combined dataset.

The DFA of phocine limb bones struggled to assign specimens to ecomorphotype categories. Jackknife validation supported the femora dataset as the most reliable model (0.77), but there was substantial overlap between all categories in LD1 of this model (Fig. 2B). In the humeri and combined datasets, there was some separation between Ecomorphotypes 1, 3 and 4 along LD1, but no separation for Ecomorphotype 2, except between Ecomorphotype 2 and 1 in the combined dataset (Fig. 2). Humeri and femora specimens of the fossil taxon *Phocanella pumila* were not consistently assigned to the same ecomorphotype, and were assigned to all ecomorphs except Ecomorphotype 3 (Table 1). For the fossil specimens assigned an ecomorphotype by *Koretsky, Alexander & Rahmat (2020)* (*Leptophoca proxima*, Ecomorphotype 3; *Monachopsis pontica*, Ecomorphotype 3, *Praepusa pannonica*, Ecomorphotype 1; *Praepusa vindobonensis*, Ecomorphotype 3; *Pontophoca sarmatica*, Ecomorphotype's 3 and 5) none were assigned by the DFA to their 'correct' ecomorphotype, as all were assigned to Ecomorphotype 2 with the exception of *Pontophoca sarmatica* (Ecomorphotype 4) (Table 1).

**Table 1 Posterior probabilities of assignment of fossils to ecomorphotype categories by the discriminant function analysis.** Fossil phocine humeri and femora represent isolated specimens and were analysed separately.

| Taxon | Specimen and element | Ecomorphotype 1 | Ecomorphotype 2 | Ecomorphotype 3 | Ecomorphotype 4 |
|---|---|---|---|---|---|
| *Phocanella pumila* | USNM 329059 (humerus) | 0.04% | 99.16% | 0.81% | 0.002% |
| | USNM 171151 (humerus) | 99.87% | 0.11% | 0.02% | 0.002% |
| | NHMUK PV M1199 (cast of IRSNB 1080-M227, humerus) | <0.001% | 99.96% | 0.03% | 0.02% |
| | NHMUK PV M1206 (cast of IRSNB 1101-M234, humerus) | <0.001% | 97.31% | 2.69% | <0.001% |
| | USNM 305283 (femur) | <0.001% | 99.63% | <0.001% | 0.37% |
| | USNM 329060 (femur) | 1.59% | 0.43% | <0.001% | 55.25% |
| | USNM 175217 (femur) | <0.001% | 76.11% | <0.001% | 23.88% |
| | USNM 181649 (femur) | <0.001% | 94.27% | 0.02% | 5.53% |
| | USNM 481569 (femur) | 96.08% | 0.55% | <0.001% | 3.36% |
| *Nanophoca vitulinoides* | NHMUK PV M1212 (cast of IRSNB 1063-M242, humerus) | <0.001% | 96.99% | 0.14% | 2.87% |
| | NHMUK PV M1216 (cast of IRSNB 1049-M247, femur) | 34.15% | 23.42% | <0.001% | 42.43% |
| *Cryptophoca maeotica* | USNM 214979 (femur, cast of LPB 259) | 0.004% | 88.25% | 0.3% | 11.45% |
| *Leptophoca "amphiatlantica"* | USNM 321926 (femur) | <0.001% | 99.99% | <0.001% | <0.001% |
| *Leptophoca proxima* | USNM 559330 (femur) | 0.01% | 97.68% | 0.003% | 2.31% |
| *Monachopsis pontica* | USNM 214967 (femur, cast of LPB 21) | <0.001% | 99% | <0.001% | 0.99% |
| *Praepusa vindobonensis* | USNM 214993 (femur, cast of LPB 158?) | <0.001% | 93.03% | <0.001% | 6.97% |
| *Praepusa? pannonica* | USNM 214978 (femur, cast of LPB 5?) | <0.001% | 99.7% | <0.001% | 0.3% |

# DISCUSSION

## The 'Ecomorphotype Hypothesis' is not valid

A large portion of the phocid fossil record has been referred to existing or newly erected taxa using the 'Ecomorphotype Hypothesis' (*Koretsky, 2001*; *Valenzuela-Toro & Pyenson, 2019*; *Koretsky, Alexander & Rahmat, 2020*), but until now this hypothesis had never been quantitatively assessed. Our results clearly demonstrate that there is no quantitative basis or support for this hypothesis.

Firstly, the results of the PCAs indicate substantial overlap in the morphospaces between ecomorphotypes (Fig. 1). At least some degree of separation between groups should be expected along at least one PC axis if ecomorphotypes were morphologically characteristic groups. But this is not the case based on our results (Fig. 1).

Secondly, the DFA indicates that ecomorphotypes poorly perform as discriminating morphological/ecological groupings (Fig. 2). The best performing model by Jackknife validation (femur only) has the most overlap between the ecomorphotype categories (Fig. 2B). For the next best model (humerus only), Ecomorphotype 2 overlaps with all other ecomorphotypes (Fig. 2A). This is likely due to Ecomorphotype 2 possessing the

most species compared to the other ecomorphotypes (*Koretsky, Alexander & Rahmat, 2020*). The combined dataset does seem to demonstrate some discrimination between ecomorphotypes, but is not supported after jackknife validation. In addition, it would not function well as a model, as it would require testing possibly countless combinations of unknown fossil humeri and femora from a given formation. For some formations, such as the Yorktown Formation and Calvert Formation, which have thousands of unassociated phocid fossils (*Valenzuela-Toro & Pyenson, 2019*) this is not feasible.

Thirdly, the DFA failed to assign any of the fossil taxa (treated as unknowns in our analysis; Table 1) to their previously assigned ecomorphotypes (*Koretsky, Alexander & Rahmat, 2020*). In the case of *Phocanella pumila*, for which the *Churchill & Uhen (2019)* dataset contained multiple humeri and femora specimens, the DFAs failed to assign them to a consistent ecomorphotype. This would cast particular doubt on not only the referral of bulk fossil material using the 'Ecomorphotype Hypothesis', but also on proposed fossil-only ecomorphotypes (*e.g.*, Ecomorphotype 5 for Phocinae; *Koretsky, Alexander & Rahmat, 2020*).

The above results affirm concerns on the 'Ecomorphotype Hypothesis' by recent articles (*Dewaele, Lambert & Louwye, 2017a*; *Dewaele et al., 2018*; *Churchill & Uhen, 2019*; *Valenzuela-Toro & Pyenson, 2019*; *Rule et al., 2020a*). We therefore recommend that, going forward, the 'Ecomorphotype Hypothesis' be abandoned as a method of referring otherwise unassociated fossil material to the same species. Instead, future rigorous taxonomy should employ a more systematic approach which involves demonstrating overlap of bony elements between associated and isolated specimens. This is a much more reliable approach, and has been employed recently as more complete fossil phocid material becomes available (*Amson & de Muizon, 2014*; *Valenzuela-Toro et al., 2016*; *Dewaele et al., 2018*; *Rule et al., 2020a, 2020b*; *Dewaele & de Muizon, 2024*).

## Implications for phylogenetic and macroevolutionary inferences

The misidentification of phocid fossil occurrences has important implications for studies that aim at inferring phylogenetic and diversification patterns in this group. Modern methods such as PyRate (*Silvestro et al., 2014*; *Silvestro, Salamin & Schnitzler, 2014*) rely on multiple occurrences for each species to properly estimate the preservation rates, and consequently "true" times of speciation and extinction. When fossil specimens are incorrectly assigned to a given species, this would therefore impact all the estimations, and will hence provide incorrect information about the diversification dynamics of the group. Based on our results, it is possible to say that we would not be able to use the fossil record for phocids to estimate the diversification dynamics of phocids, given that apart from very few cases most species have too few occurrences (in most cases only one per species; *Valenzuela-Toro & Pyenson, 2019*; *Berta, Churchill & Boessenecker, 2022*) to be suitable to be used in the analyses. Additionally, a recently developed method that allows for the estimation of phylogenetic trees including both extant and fossil species (metatrees, *Lloyd et al., 2016*) would also be impacted by this misassignment. This method draws information from the age range of fossil species to estimate the branch lengths related to those lineages. With fossil occurrences wrongly assigned to a given species, it can show a

much longer range than what we would get by only considering the correctly assigned ones (or even only the type specimen, using the geological stage in which the specimen was deposited as a surrogate age).

## The validity of fossil phocid species

*Valenzuela-Toro & Pyenson (2019)*, in their analysis of the fossil record of pinnipeds, highlighted the need for best taxonomic practices in the field, as these have downstream effects on phylogenetic and macroevolutionary analyses of pinnipeds. Recent reviews of the fossil record of phocids have highlighted the need for revision of problematic taxa, especially phocines (*Berta, Churchill & Boessenecker, 2018*; *Berta, Churchill & Boessenecker, 2022*). *Churchill & Uhen (2019)* suggested that, based on their morphometric analysis of phocid limb bones, that fossil phocids known from described isolated limb elements should be considered *nomina dubia*. Whilst one study has so far reviewed two of these taxa (*Rule et al., 2020a*), the vast majority remain unassessed. The results presented here demonstrate that all fossil phocid species whose hypodigms have been expanded using the 'Ecomorphotype Hypothesis' should instead be restricted to the type specimens only.

Considering the above, we assert that all fossil phocids described using isolated limb bones be considered *nomina dubia* (Table 2), which represents over 50% of previously described extinct phocid taxa (Fig. 3). In addition, other taxa described from equally incomplete and isolated material (*e.g., Lobodon vetus* and *Palmidophoca callirhoe*, whose type specimens are isolated dentitions) should also be considered *nomina dubia*. Given that most of the species now considered *nomina dubia* are from the Northern Hemisphere and dated to the Miocene (Fig. 4), especially the eastern North Atlantic, this taxonomic upheaval has serious implications for the evolution of Phocidae. This region is often invoked as the area where Phocidae originated and diversified (*Berta, Churchill & Boessenecker, 2018*; *Rule et al., 2020b*; *Park et al., 2024*), and includes the entirety of the fossil record of the northern phocids (Phocinae). As such, any future research efforts to uncover, describe, and name more complete fossil phocids should also focus on the Miocene of Europe.

We also recommend the revision of several taxa. Specifically, *Phocanella pumila* is described from isolated axial postcrania, and the taxonomic utility of these elements has not been tested. Other more complete taxa might also need revision, such as *Devinophoca emryi*. Despite the type specimen being a skull, this taxon is known from the same locality and unnamed formation (underlying the Studienka Formation) as the only other member of its genus (*Devinophoca claytoni*), and both are morphologically very similar and usually end up close to each other on phylogenies (*e.g., Rule et al., 2020b*; *Park et al., 2024*). Another taxon, *Histriophoca alekseevi*, is likely valid as it is represented by a partial skull, however the referral to the modern genus *Histriophoca* should be revisited in light of recent work. Another specimen that may need to be revisited is *Australophoca changorum*, the type specimen of which is a fragmentary humerus, radius, and ulna, and the paratype of which includes various forelimb and hindlimb bones (*Valenzuela-Toro et al., 2016*). While

**Table 2 Taxonomic review of fossil Phocidae.** Taxa listed do not include any taxa or names that have fallen out of use in the literature. Comments either reflect reasoning for status, or a citation in support of status. List is expanded from *Berta, Churchill & Boessenecker (2022)*.

| Taxon | Status | Comments |
|---|---|---|
| *Acrophoca longirostris* | Valid | Type is a partial skeleton |
| *Afrophoca libyca* | Needs revision | Type is a fragmentary mandible |
| *Auroraphoca atlantica* | *Nomen dubium* | *Churchill & Uhen (2019)* analysis |
| *Australophoca changorum* | Valid, needs revision | Type is a fragmentary forelimb, but more complete undescribed fossils exist |
| *Batavipusa neerlandica* | *Nomen dubium* | *Churchill & Uhen (2019)* analysis |
| *Callophoca obscura* | *Nomen dubium* | *Berta et al. (2015)* and *Churchill & Uhen (2019)* analysis, supported by *Rule et al. (2020a)* |
| *Cryptophoca maeotica* | *Nomen dubium* | *Churchill & Uhen (2019)* analysis |
| *Devinophoca claytoni* | Valid | Type is a partial skull |
| *Devinophoca emryi* | Needs revision | Similar to *D. claytoni*, and from same formation as *D. claytoni* |
| *Eomonachus belegaerensis* | Valid | Type is a near complete skull |
| *Frisiphoca aberratum* | *Nomen dubium* | *Churchill & Uhen (2019)* analysis |
| *Frisiphoca affine* | *Nomen dubium* | *Churchill & Uhen (2019)* analysis |
| *Gryphoca nordica* | *Nomen dubium* | *Churchill & Uhen (2019)* analysis |
| *Gryphoca similis* | *Nomen dubium* | *Churchill & Uhen (2019)* analysis |
| *Hadrokirus martini* | Valid | Type is a partial skeleton |
| *Hadrokirus novotini* | Needs revision | Type is a fragmentary mandible |
| *Histriophoca alekseevi* | Valid, needs revision | Type is a partial skull. Needs revision for referral to modern genus *Histriophoca* |
| *Homiphoca capensis* | Valid | Type is a partial skull |
| *Homiphoca murfreesi* | Needs revision | Type is a fragmentary mandible |
| *Kawas benegasorum* | Valid | Type is a partial skeleton |
| *Leptophoca proxima* | Needs revision | Type is a humerus, but undescribed more complete fossils exist. |
| *Leptophoca "amphiatlantica"* | *Nomen dubium* | *Dewaele, Lambert & Louwye (2017a)* |
| *Lobodon vetus* | *Nomen dubium* | Type is an isolated postcanine, likely represents a modern specimen |
| *Magnotherium johnsii* | Valid, needs revision | Type is a fragmentary skull, but not character rich |
| *Mesotaria ambigua* | *Nomen dubium* | *Berta et al. (2015)* and *Churchill & Uhen (2019)* analysis, supported by *Rule et al. (2020a)* |
| *Messiphoca mauretanica* | Valid, needs revision | Type is a partial skeleton, skull is referred |
| *Miophoca vetusta* | *Nomen dubium* | *Dewaele, Lambert & Louwye (2017a)* |
| *Monachopsis pontica* | *Nomen dubium* | *Churchill & Uhen (2019)* analysis |
| *Monotherium delognii* | *Nomen dubium* | *Berta et al. (2015)* |
| *Monotherium? wymani* | Needs revision | Type is an ear region |
| *Nanophoca vitulinoides* | Valid | Type is a partial skeleton |
| *Noriphoca gaudini* | Valid | Type is a partial skull |
| *Pachyphoca chapskii* | *Nomen dubium* | *Churchill & Uhen (2019)* analysis |
| *Pachyphoca ukrainica* | *Nomen dubium* | *Churchill & Uhen (2019)* analysis |
| *Pachyphoca volkodavi* | *Nomen dubium* | *Churchill & Uhen (2019)* analysis |
| *Palmidophoca callirhoe* | *Nomen dubium* | Type is a tooth |
| *Phoca bessarabica* | *Nomen dubium* | *Churchill & Uhen (2019)* analysis |
| *Phoca moori* | *Nomen dubium* | *Churchill & Uhen (2019)* analysis |
| *Phocanella pumila* | Needs revision | Type is an isolated atlas |
| *Piscophoca pacifica* | Valid | Type is a partial skeleton |
| *Planopusa semenovi* | Valid | Type is partial snout |

| Taxon | Status | Comments |
| --- | --- | --- |
| *Platyphoca danica* | *Nomen dubium* | *Churchill & Uhen (2019)* analysis |
| *Platyphoca nystii* | *Nomen dubium* | *Berta et al. (2015)* |
| *Platyphoca vulgaris* | *Nomen dubium* | *Churchill & Uhen (2019)* analysis |
| *Pliophoca etrusca* | Valid | Type is a partial skeleton |
| *Pontophoca jutlandica* | *Nomen dubium* | *Churchill & Uhen (2019)* analysis |
| *Pontophoca sarmatica* | *Nomen dubium* | *Churchill & Uhen (2019)* analysis |
| *Pontophoca simionescui* | Needs revision | No designated type specimen |
| *Praepusa boeska* | *Nomen dubium* | *Churchill & Uhen (2019)* analysis |
| *Praepusa magyaricus* | *Nomen dubium* | *Churchill & Uhen (2019)* analysis |
| *Praepusa pannonica* | Needs revision | Type is a mandible |
| *Praepusa procaspica* | Valid, needs revision | Type is associated forelimb |
| *Praepusa tarchankutica* | Valid, needs revision | Type is a cranium |
| *Praepusa vindobonensis* | *Nomen dubium* | *Churchill & Uhen (2019)* analysis |
| *Pristiphoca occitana* | *Nomen nudum* | *Berta et al. (2015)* |
| *Properiptychus argentinus* | Needs revision | Type is a partial maxilla |
| *Prophoca rousseaui* | *Nomen dubium* | *Churchill & Uhen (2019)* analysis |
| *Sarcodectes magnus* | Valid | Holotype is a partial skull |
| *Sarmatonectes sintsovi* | *Nomen dubium* | *Churchill & Uhen (2019)* analysis |
| *Seronectes meherrinensis* | *Nomen dubium* | Type is a fragmentary innominate |
| *Terranectes magnus* | *Nomen dubium* | *Dewaele et al. (2018)* |
| *Terranectes parvus* | *Nomen dubium* | *Dewaele et al. (2018)* |
| *Virginiaphoca magurai* | *Nomen dubium* | *Churchill & Uhen (2019)* analysis |

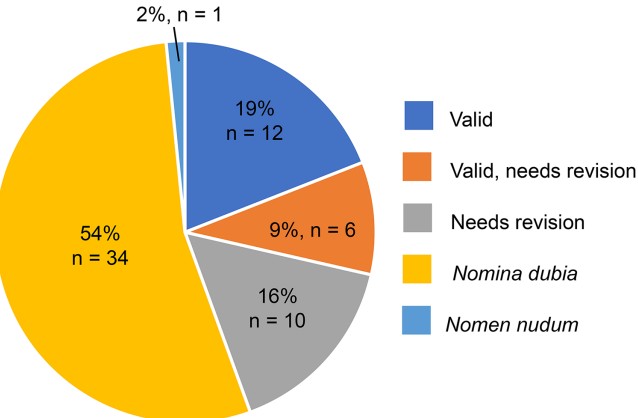

**Figure 3 Status of the taxonomic validity of fossil phocid species.** Distribution of data from Table 2. "*n*" = number.

this specimen is technically known from associated material, the state of preservation of those fossils means that its utility as a type specimen could be revisited.

The genus *Praepusa*, from the middle Miocene-early Pliocene of Europe, also needs to be revised. While the type species *Praepusa pannonica* (which has a holotype consisting of

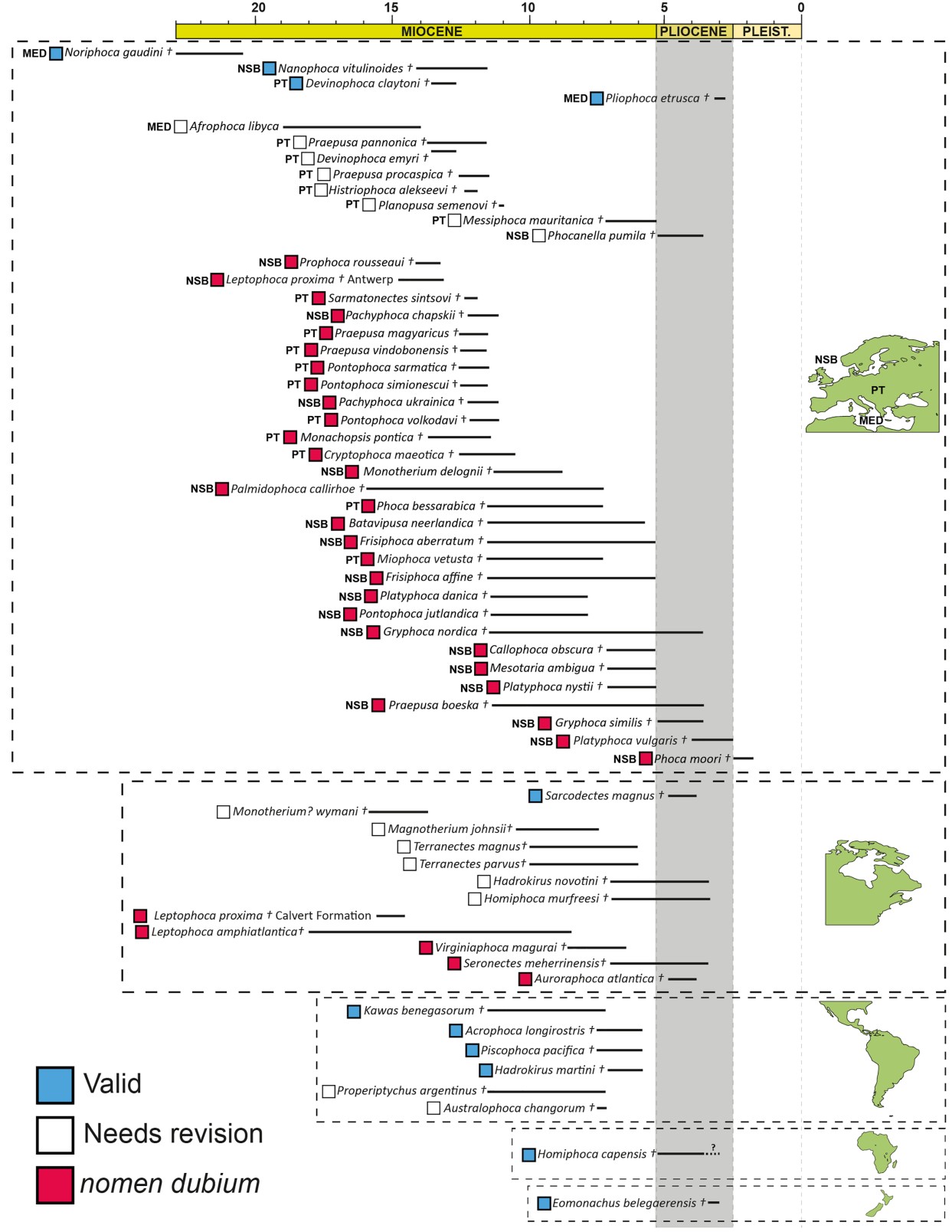

**Figure 4 Geographic and temporal distribution of the revised fossil record of Phocidae.** Stratigraphic ranges taken from the Palaeobiology database. NSB = North Sea Basin; PT = Paratethys; MED = Mediterranean. *Lobodon vetus*, *Praepusa tarchankutica*, and *Pristiphoca occitana* excluded due to lack of provenance or stratigraphic information.

a fragmentary mandible with teeth), might be valid, the referral of *Praepusa procaspica*, *Praepusa trachankutica*, and *Praepusa vindobonensis* to the same genus is questionable as none of the type specimens overlap. Some of these species either have craniomandibular material as type specimens (*Praepusa pannonica* and *Praepusa trachankutica*) or have had complete crania referred to them in the past (*Praepusa vindobonensis*). As such, it is critical that these particular specimens are revisited, as they likely will be beneficial to investigating the systematics of Phocinae, and will be critical to any phylogenetic analysis of the group.

We hope that the results of this and other recent work (*Churchill & Uhen, 2019*; *Valenzuela-Toro & Pyenson, 2019*) provokes new discussion and research into the minimum requirements for the quality of holotype specimens in fossil Phocidae. We recommend that in future, any novel method for aiding the referral of phocid fossils (or indeed fossils from any group) be grounded in a rigorous, quantitative, and statistically sound framework. This will help avoid taxonomic ambiguity in the fossil record, and ensure that any subsequent macroevolutionary analyses are based on solid taxonomic foundations.

## CONCLUSIONS

We tested the proposed 'Ecomorphotype Hypothesis', a qualitative system for the referral of phocid fossils, using quantitative methods. Our analyses do not support the 'Ecomorphotype Hypothesis', and found that ecomorphotype groupings performed poorly for assigning unknown fossils. We therefore find that fossils referred to taxa using this method cannot be linked to isolated and non-overlapping type specimens. As a result of our findings, and those from previous studies, we find that the majority of extinct phocid species should be considered *nomina dubia*.

## ACKNOWLEDGEMENTS

Thanks to R-P Miguez, P. Kokkini, N. Adams (Natural History Museum, London) and N. Kohno and Y. Tajima (National Museum of Nature and Science, Tokyo) for access to specimens in their care. We thank the editor (C. Stefen) and three reviewers (L. Dewaele, A. Valenzuela-Toro, A. Berta) for their comments that improved the quality of the manuscript.

### Funding

James Patrick Rule was funded by a UKRI Fellowship (Grant No. EP/X021238/1) from the Engineering and Physical Sciences Research Council. Gustavo Burin and Travis Park were funded by a Leverhulme Trust Research Project grant (RPG-2019-323). Travis Park was also funded by an Australian Research Council DECRA Fellowship (DE220101296). The funders had no role in study design, data collection and analysis, decision to publish, or preparation of the manuscript.

## Grant Disclosures

The following grant information was disclosed by the authors:
UKRI Fellowship: EP/X021238/1.
Engineering and Physical Sciences Research Council.
Leverhulme Trust Research: RPG-2019-323.
Australian Research Council DECRA Fellowship: DE220101296.

## Competing Interests

The authors declare that they have no competing interests.

## Author Contributions

- James Patrick Rule conceived and designed the experiments, performed the experiments, analyzed the data, prepared figures and/or tables, authored or reviewed drafts of the article, and approved the final draft.
- Gustavo Burin performed the experiments, authored or reviewed drafts of the article, and approved the final draft.
- Travis Park performed the experiments, authored or reviewed drafts of the article, and approved the final draft.

## Data Availability

The raw data, input files, and code for this article are available at Figshare: Rule, James (2024). Ecomorphotype R code and input files. figshare. Dataset. https://doi.org/10.6084/m9.figshare.25029554.v1.

## Supplemental Information

Supplemental information for this article can be found online at http://dx.doi.org/10.7717/peerj.17592#supplemental-information.

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
