# Peer review of "A quantitative test of the “Ecomorphotype Hypothesis” for fossil true seals (Family Phocidae)"

_PeerJ, doi:10.7717/peerj.17592_

## Round 0.1 · original submission · Minor Revisions

Dear Authors,

The work is interesting and worth publishing and I have only a few points that could be clarified.

Figure 1 - indicate how many specimen are in the different ecomorphotype categories and fossils (add n = … )
for © be more specific : “humeri and femora together.
And please explain why that makes sense; these are different bones and have different functions even in seals, so I am a skeptical about combining both. Is it because Koretsky also did it when developing the method??.
And explain the % at the axis

Figure 2 – explain “proportions of traced by … (certainly not clear to every reader); same with jackknife validation.
And maybe comment on the apparent separation of ectomorphotype 1 here.
Table 1 – there are two headings. Please make it clear that it is your analysis of the probability of ectomorph categories and whether humeri and femora were used together.

Text
Even though you refer to the subset published in Curchrill & Uhen, it would be useful to have the exact dataset of your study in an additional supplementary file with the additional material you used. And it would be useful to have not just the collection numbers like in Churchill & Uhen, but also which bone it is. And the measurements of the additional 36 specimen you used.
And I miss a list of the a priori grouping of the used material into the ecomorphotypes which would then also allow us to see how many specimens are in each category – so please add such an a priori list!!
And as reviewer 2 indicates also add some information on these categories in the text.
Add a thank you to the reviewers.

Please consider the comments of the reviewers as well and I am looking forward to the revised manuscript.

·

Basic reporting

To the editor,
To the reviewers,

To start, I want to thank the authors for the study they carried out. The ecomorphotype hypothesis has indeed been used by certain colleagues in the field to justify the otherwise poor selection of type specimens to create new fossil genera and species of phocid seals, because from its conception, the ecomorphotype hypothesis had little to no quantitative data in support of it.

The manuscript is concise, yet still very well structured and to the point. The references are adequate. However, at some points, it may need a few additional lines to make things more clear. More on that further down in the review report.
Based on my review of the manuscript, and the few comments I have, I suggest minor revisions, primarily also to make the authors reflect on a -literally- debatable issue, but I am glad to accept if the reviewers would ignore these.
I return the annotated text with very minor typographical issues.

Experimental design

All criteria for the experimental design are met, with the possible exception of the slight lack of details to replicate the analyses easily: in addition to the specimen list in the supplementary material, I suggest to add the measurements. This allows others to re-run the tests much more easily.
However, this is a minor issue and can be resolved easily.

Validity of the findings

The findings quantify what has been expected by many in our field. The data in the main text is concise and what needed to be there. However, I think that a few more details could be provided as supplementary information: graphs, statistics, etc.

Another issue is the table (re-)assessing the different fossil species. For personal discussions in the past, the lead author is well aware that I applaud the sceptical approach and that many fossil seal taxa should be treated as nomina dubia. However, I think that this table is slightly outside the scope of the research question and the results, which focus on the ecomorphotype hypothesis. Also, it tars all taxa with the same brush. I agree that long bones as type specimens are very little useful and I totally agree with considering many nomina dubia, including some that I named. However, there is a grey zone between what is a good type specimen and what is an absolutely bad type specimen. The example that I use in the text is the comparison between Batavipusa and Australophoca. Batavipusa is based on a complete humerus, whereas Australophoca is based on a severely battered humerus, plus radius and ulna. In this manuscript, the former is automatically considered a nomen dubium, whereas the latter is valid (without recommendation to revise it). In my opinion, arguments can be made in favor of and against both designations.
Therefore, I would suggest the authors to be cautious here. I suggest to be more careful and talk more generalistically here, or elaborate more about their reasoning. As it is now, I get the impression that the dismissal of isolated humeri and femora is based on Churchill and Uhen (2019), which I perfectly get, but a battered humerus with only two other battered bones is considered valid, despite their state? I feel that this might open a can of worms if no more information explaining the authors' reasoning is provided.

However, the rest, i.e. core, of the work is sound and this should only require minor revisions.

I look forward to see this manuscript being published soon!

·

Basic reporting

I am very pleased with the quality of this study. It is well-written and logically organized. The study addresses an important practice in pinniped paleontology, and its results have broad implications for our understanding of their evolution, including diversity estimates and paleobiogeographic inferences. The introduction provides a good overview of the Ecomorphological Hypothesis and its shortcomings. The methods are appropriate and the conclusions are supported by the results. My only major recommendation is to report the original and published raw data used in this study and attend to a few comments described below.

Experimental design

No comment.

Validity of the findings

No comment.

Additional comments

Dear authors
Congratulations on this much-needed study. Some forms of the "Ecomorphotype Hypothesis" have been around for more than two decades (Koresty, 2001) and have been used to support the artificial creation of several fossil species from isolated and fragmentary skeletal remains. The implications of your study are profound and should encourage a new generation of more critical examinations of the seal fossil record, especially from the Northern Hemisphere. The manuscript is well-articulated and the results are sound. I have only a few comments to enable a more thorough understanding of the Ecomorphotype Hypothesis by nonspecialists, and from there to strengthen and contextualize your findings.

1. As someone who studies fossil pinnipeds, I know about the Ecomorphotype Hypothesis and the five ecomorphotypes established by Koretsky (2020). However, I imagine that most readers are not familiar with these classification schemes. Consequently, it would be convenient to add a section describing in more detail these ecomorphotypes in the Introduction or Materials and Methods. I encourage authors to be explicit about the characters and traits that define each of these groups, their putative ecological attributes, and the species assigned to each of them.

2. I suggest adding a statement describing the expected results if the Ecomorphotype Hypothesis is validated (e.g., we would expect a clear separation in the morphospace in this or that direction) or not. There is some of this in the discussion between lines 153 and 156, but I think this information should be earlier in the text so that non-specialist readers know what to expect in each case. I know this sounds a bit obvious but it is important to always be explicit about the scenarios when testing hypotheses.

3. Report the raw data.

Other very minor comments/suggestions are described below:
Line 19. The "ecomorphotype hypothesis" should begin with a capital E or H. Moreover, the typography for this should be revised and made consistent throughout the text. For instance, some versions include a capital E for ecomorphotype, while others do not.

Line 83. Did you include females and males? Does it matter to have both sexes represented?

Line 88. The species name should be in italics. I also suggest including the common name when you first mention an extant species.

Line 95. Did a single researcher take all the original measurements?

Lines 124-126. How does the morphology vary along the PC1 and PC2? What is the direction of morphological variation for each of the principal components?

Lines 220-222. Given these remarkable results, could you elaborate on their implications for our understanding of phocid evolution from a geographic & temporal perspective? I see that there is a strong trend towards Miocene northern seals, with most of these seals becoming invalid, but it would be fantastic if you could be more specific and highlight the need to pay more attention to the study of such and such pinniped assemblages, formations, time periods, etc.

Line 229. From which formation was Devinophoca emryi described?

Lines 232-236. Like above, please add more information about the geographic and/or temporal origin of these taxa.

Again, this is a great manuscript, and I look forward to seeing it published!

Ana Valenzuela-Toro

·

Basic reporting

It’s taken 20+years…. but finally the ecomophological hypothesis applied to fossil phocids has been rigorously tested and found to be invalid. Over the years several other scientists have suggested that this method that relies on subjective morphological criteria used to refer isolated fossil phocid postcrania to taxa is problematic. For the first time the authors have used quantitative methods to clearly show substantial overlap in ecomorphologic types. They recommend that more than 50% of named fossil phocids diagnosed using the ecomorphological hypothesis be declared nomen dubia. I completely agree. It is time for the taxonomy of fossil phocids to be re-examined and revised where necessary. This paper provides the framework for doing so and brings phocid taxonomy into the 21st century. I have no suggestions for improvement. The paper is logically organized, with appropriate methods and well explained analyses.

Experimental design

see above

Validity of the findings

see above

---

## Round 0.2 · Minor Revisions

Dear authors,

There are still some concerns concerning the manuscript that should be taken care of before it can be accepted.

In addition, the Section Editor Kenneth De Baets has commented that:

"I feel the science is good and figures. However, because it as it is the main part of the study that figure showing the definition of taken measurements/parameters should ideally be shown in the manuscript for the sake of reproducibility for those not so familiar with them (i feel referring to another publication even if open access is insufficient) or at minimum in supplementary material. It could be a an extra figure or added to some of the figures already existing. Also the way those measurements were taken should be elaborated upon in text (physical specimens, illustrations, 3D model). In some cases, the references are incomplete (e.g., Churchill and Uhen 2019 need page numbers and/or doi)."

Please add the recommended information.

Yours
Clara Stefen

·

Basic reporting

Reporting on a second review of a manuscript that I gave very minor revisions during the first reading. Consequently, this report will be very concise.
My main concerns in the previous reviewing were the reproducibility of the analyses, as well as a few issues with the delineation of what bones as type specimens make a species either valid or a nomen dubium.
The criteria for basic reporting are met.

Experimental design

Criteria for the experimental design are met.

Validity of the findings

As noted above, my main concerns during the previous reviewing round are addressed. I am happy to see that the analyses by the authors can now be reproduced much more easily.

I do have to say that I am not quite convinced with black-and-white delineation of what isolated (or few associated) bones and bits of bones can be used as type specimens.
In my opinion, there is a great gray zone of isolated humeri and femora that are highly questionable as type specimens, but there are a few that seem so outlandishly unique, that there is no doubt that they cannot be confused with any other species. Even though we all agree that isolated long bones should no longer be used as type specimens, ever since Churchill and Uhen published their 2019 paper.
However, I do think that there is another time and place for the authors and me to have a chat about this and maybe work on a few guidelines for our community.

I look forward to see this manuscript turn into a published paper, finally invalidating the ecomorphotype hypothesis.

·

Basic reporting

I have reviewed this new version of the manuscript. I am pleased that the authors have taken my comments and those of other reviewers into account. Although the original manuscript was already complete, this new version discusses the classification schemes of the Ecomorphotype Hypothesis and the implications of the study in more detail, making it more accessible and useful to non-pinniped specialists.
I look forward to seeing this manuscript published!

Experimental design

No comment.

Validity of the findings

No comment.

Additional comments

No comment.

---

## Round 0.3 · accepted · Accept

Thanks for the revision. It includes a good figure illustrating the measurements in the supplementary material and this is I think sufficient.